# Deterioration of Retinal Blood Flow Parameters in Branch Retinal Vein Occlusion Measured by Doppler Optical Coherence Tomography Flowmeter

**DOI:** 10.3390/jcm9061847

**Published:** 2020-06-13

**Authors:** Kengo Takahashi, Youngseok Song, Kenji Sogawa, Takafumi Yoshioka, Tomofumi Tani, Satoshi Ishiko, Akitoshi Yoshida

**Affiliations:** Department of Ophthalmology, School of Medicine, Asahikawa Medical University, Asahikawa 078-8510, Japan; kt@asahikawa-med.ac.jp (K.T.); cdh72740@yahoo.co.jp (K.S.); tkfmysok@gmail.com (T.Y.); tonytani555@gmail.com (T.T.); ishiko@asahikawa-med.ac.jp (S.I.); ayoshida@asahikawa-med.ac.jp (A.Y.)

**Keywords:** Doppler optical coherence tomography, branch retinal vein occlusion, retinal blood flow

## Abstract

Background: Doppler optical coherence tomography (DOCT) flowmeter can be used to measure retinal blood flow (RBF) parameters, including vessel diameter, blood velocity, and the absolute value of RBF within 2.0 s. We investigated the RBF parameters in eyes with branch retinal vein occlusion (BRVO) using a DOCT flowmeter. Methods: Seventeen patients with unilateral BRVO were enrolled. All subjects underwent comprehensive ophthalmologic examinations. The RBF parameters were assessed from three veins, i.e., (1) an occluded vein, (2) a non-occluded vein in the BRVO eyes, and (3) an equivalent (superior or inferior) vein in the fellow eye (non-affected vein), using prototype DOCT flowmeter (Topcon, Tokyo, Japan). Moreover, the correlation between RBF parameters and the best corrected visual acuity (BCVA) was examined. We investigated the correlation between (1) the RBF parameters and the time from the initial visit, (2) the RBF parameters and the time from the last injection, and (3) the RBF parameters and the number of anti-vascular endothelial growth factor injections (VEGF). Results: The diameter of the occluded vein (95.9 ± 24.7 µm) was smaller than that of the non-occluded vein (127.9 ± 23.7 µm) and that of the healthy veins (116.4 ± 13.9 µm). The RBF was lower in the occluded veins (4.7 ± 3.7 µL/min) than that in the non-occluded veins (10.3 ± 5.1 µL/min; *p* < 0.01) and in the fellow eyes (8.6 ± 4.0 µL/min; *p* = 0.013). In contrast, the blood velocity was not significantly different among the three types of veins. BCVA was correlated with the diameter of the occluded vein (ρ = 0.711, *p* = 0.001) but not with the RBF and blood velocity. The time from the initial visit, the time from the last injection, and the total number of anti-VEGF injections were not associated with any RBF parameters on the occluded vein. Conclusions: The RBF was significantly lower in the occluded veins than that in the other veins, and the diameter of the occluded vein was significantly smaller than that of the other veins in patients with BRVO. However, neither the time from the initial visit, nor the time from the last injection, nor the number of anti-VEGF injections were correlated with the RBF parameters on the occluded vein.

## 1. Introduction

Retinal vein occlusion (RVO) is the second most common circulatory disease, and branch RVO (BRVO) represents most the cases of RVO. [1] Previous studies have reported the prognostic factors of BRVO, which include aging [2] and large areas of non-perfusion [3,4]. Moreover, the presence of macular ischemia is an important factor for the final visual acuity (VA) outcomes in BRVO [3,4]. Concomitantly, another study [5] reported that additional photocoagulation treatment for macular ischemia together with anti-vascular endothelium growth factor (VEGF) therapy did not improve final VA outcomes. Although ischemic conditions are an important factor in the pathogenesis of BRVO, the involvement of retinal blood flow (RBF) reduction in the pathogenesis of BRVO remains unclear.

Several clinical studies have suggested that the reduction in the retinal blood velocity measured by various devices could be related to the pathophysiology of BRVO [6,7,8,9]. However, no study has investigated the changes in the RBF of the occluded vein in eyes with BRVO.

Previously, we developed a Doppler optical coherence tomography (DOCT) flowmeter method by modifying spectral-domain optical coherence tomography (SD-OCT) system (3D OCT-1 Maestro, Topcon Corp., Tokyo, Japan), which can measure retinal arterial and vein blood velocity noninvasively, as blood velocity waveforms [10].

The aims of the study were (1) to assess the characteristics of RBF as measured by DOCT flowmeter on the occluded vein in eyes with BRVO and (2) to evaluate the correlation between the RBF and best corrected visual acuity (BCVA) in eyes with BRVO.

## 2. Material and Methods

The ethics committee of Asahikawa Medical University (Asahikawa Medical University Independent Ethics Committee) approved the study protocol (approval number, 17114), which adhered to the tenets of the Declaration of Helsinki. All patients provided written informed consent.

### 2.1. Inclusion/Exclusion Criteria and Diagnosis of BRVO

Seventeen consecutive patients with BRVO (six months or later from the initial visit) were recruited from 1 December 2017 through to 31 March 2019 at the medical retina clinic of Asahikawa Medical University. Two retina specialists (KS and KT) established the diagnosis of BRVO with the following diagnostic criteria: the presentation of feather or flame-shaped hemorrhage on fundus due to an occlusion of branch retinal veins. All patients received one initial intravitreal injection (IVI), and patients with persistent macular edema (ME) at subsequent follow-up visits received additional intravitreal anti-VEGF injection following a pro re nata (PRN) regimen. Patients with glaucoma, diabetes, fundus abnormalities not caused by BRVO, pre-existing retinal dystrophies, uncontrolled hypertension (blood pressure > 160/100 mmHg), and any history and evidence of past events or the presence of hypertensive retinopathy or vascular distress were excluded. We also excluded the patients who underwent the anti-VEGF injection due to ME on the day of the DOCT flowmeter measurement. Eyes with intraocular surgeries other than cataract surgery were also excluded from the study.

### 2.2. Doppler Optical Coherence Tomography

We evaluated the vessel diameter, blood velocity, and RBF using a DOCT flowmeter according to the established method (Figure 1) [10,11,12,13]. Our system was based on a commercially available SD-OCT system (3D OCT-1 Maestro; Topcon Corp., Tokyo, Japan), which operates at the 800 nm wavelength range [12]. 

According to the previously described segmental method, the image-capturing software was upgraded for Doppler imaging, and the image-processing software was newly developed to measure the RBF [12]. One hundred and eighty datasets were captured within approximately 2 s with a 1 mm scan length. If the Doppler angle is close to 90 degrees, the amount of Doppler shift or phase is equal to 0 [14]. The measured flow rates in the artery and the vein seemed to be stable under 85 degrees, however, it was overrated when the doppler angle exceeded 86 degrees. Therefore, we excluded the cases when the Doppler angle was calculated to be more than 86 degrees.

RBF was evaluated according to the established method (Figure 1) [10,11,12]. Before the blood flow imaging, firstly, a color fundus image was captured. Secondly, the scan location, which is half a disc diameter away from the optic disc, was selected. In this procedure, the scan location was also set to confirm that the scan was perpendicular to the vessel. Thirdly, the flow velocity was calculated by the angle between the incident beam and flow direction (Doppler angle) and the amount of Doppler shift (phase image). To calculate the Doppler angle, two scans were performed before blood flow measurement. Briefly, two scans were performed parallel to each other and perpendicular to the blood vessel at a distance of Y = 100 µm. The Doppler angle was determined by identifying the vessel center by the phase image. The blood vessel can be treated as a straight line and the measurement of the systolic and diastolic phase was performed over approximately 2 s. Using the retinal motion tracking function, the scan location was adjusted accordingly for the scanning time. In most cases, at least one systolic and one diastolic phase pulsation curve could be seen for approximately 2 s. The Doppler shift image (phase image) was determined according to the established method by determining the phase difference between the two adjacent scans [10,11,12]. The vessel location and its center and diameter were identified by registering and adding the collected phase images using the automated software (Figure 1B). The flow was then determined by the vessel diameter and velocity.

### 2.3. Study Protocol

All the measurements using the DOCT flowmeter were performed by KT. After the dilation of the pupils using a 0.5% tropicamide eye drop (Santen Pharmaceutical Co., Osaka, Japan), the subjects rested for at least 5 min in a dimly lit room at a temperature of 25 °C and their blood pressure was measured in the left arm. All subjects underwent comprehensive eye examinations, such as BCVA: logMAR (a standard logarithmic visual acuity chart was used), intraocular pressure (IOP), slit-lamp examination, and optical coherence tomography (Triton, Topcon Corp., Tokyo, Japan) to measure the central macular thickness. The mean ocular perfusion pressure was calculated at 2/3 mean blood pressure (MBP)–IOP. At least 24 h before the measurements, the subjects were instructed to refrain from intaking caffeine-containing drinks, such as coffee. RBF was measured using a DOCT flowmeter on three veins, i.e., 1. an occluded vein in BRVO eyes; 2. a non-occluded vein on the opposite hemisphere of the occluded vein in the BRVO eye; and 3. a vein in an equivalent area (supero- or infero-temporal) in the fellow eyes. The RBF was measured at veins at a half disc diameter away from the optic disc.

### 2.4. Statistical Analysis

The data were analyzed using the JMP software version 14.0.0 (SAS Inc., Tokyo, Japan). All the data are expressed as the mean ± standard deviation (SD). A Steel–Dwass test was used for non-parametric multiple comparisons of the RBF parameters among the veins. Spearman’s rank correlation coefficient was calculated to examine the associations between (1) the RBF parameters and the BCVA, (2) the RBF parameters and the time from the initial visit, (3) the RBF parameters and the time from the last anti-VEGF injection, and (4) the RBF parameters and the number of anti-VEGF injections.

## 3. Results

### 3.1. Baseline Demographics

The demographics of the study participants are summarized in Table 1. The mean age (SD) was 63.1 (SD, 10.8) years and the MBP was 102.3 (SD, 13.3) mmHg. The BCVA, as assessed using logMAR values, was worse in BRVO eyes (0.06 ± 0.15) than that in the fellow eyes (−0.09 ± 0.03; *p* < 0.001). The IOP was not significantly different between the BRVO (14.9 ± 2.3 mmHg) and the fellow eyes (14.7 ± 2.1 mmHg; *p* = 0.747).

### 3.2. RBF Parameters of DOCT Flowmeter

Table 2 lists the RBF parameters of DOCT flowmeter in the occluded vein, the non-occluded vein, and the vein in the fellow eyes. The RBF in the occluded vein (4.7 ± 3.7 µL/min) was significantly lower than that in the non-occluded vein (10.3 ± 5.1 µL/min; *p* < 0.01) and in the vein in the fellow eyes (8.6 ± 4.0 µL/min; *p* = 0.013). There were no significant differences in the blood velocity among the veins (occluded vein, 9.6 ± 5.3; non-occluded vein, 12.3 ± 5.0; vein in fellow eyes, 11.7 ± 5.3 mm/s; *p* > 0.05). The diameter of the occluded vein (95.9 ± 24.7 µm) was significantly smaller than that of the non-occluded vein (127.9 ± 23.7 µm; *p* < 0.01) and that of the vein in fellow eyes (117.8 ± 14.5 µm *p* = 0.022).

### 3.3. Correlations between Retinal Blood Flow and the Ocular Parameters in the Occluded Vein

In the occluded vein, the BCVA was correlated with a vessel diameter (ρ = 0.711, *p* = 0.001) (Figure 2).

The time from the initial visit, the time from the last anti-VEGF injection and the number of anti-VEGF injections were not associated with any of the RBF parameters (Table 3).

## 4. Discussion

The current study showed that the RBF in the occluded vein was lower than that in the non-occluded and the healthy veins. The diameter of the occluded vein was smaller than that of the non-occluded and healthy veins, while there was no significant difference in the blood velocity among the veins. These results suggest that the reduction in RBF was associated with the diameter of the veins rather than with the blood velocity. In addition, the diameter of the occluded vein was correlated with the BCVA, implying that the diameter of the occluded vein in the BRVO eyes may be an important index for BRVO.

A previous study using scanning laser Doppler flowmetry reported that the RBF and blood velocity in the occluded vein were significantly lower than they were in the non-occluded and the equivalent (in fellow eye) veins in treatment-naïve BRVO [6]. Moreover, another study using scanning laser ophthalmoscopy showed that the blood velocity in the perifoveal area was correlated with BCVA [15]. However, in the current study, blood velocity in the occluded vein was comparable to that recorded in the fellow eye, possibly because the current study enrolled patients with chronic BRVO. In addition, the diameter of, but not the blood velocity in, the occluded vein was correlated with the BCVA in the current study. This discrepancy may have stemmed from the differences between studies regarding the methods and locations used to measure RBF.

Several previous studies have reported that the presence of macular ischemia is an important prognostic factor for the final VA outcomes in patients with BRVO [3,4]. Conversely, another study [5] reported that additional photocoagulation applied to patients with macular ischemia treated with anti-VEGF injection did not improve the final outcomes, indicating that other factors might affect the VA outcomes, but not blood supply to the retina. Our results imply that the diameter of the occluded vein in the BRVO eye may reflect BCVA.

Previous reports have described changes in retinal vein diameter in patients with BRVO, in whom anti-VEGF injection causes the vasoconstriction of the retinal vein [16,17,18]. Here, the number of anti-VEGF injections was not significantly correlated with the diameter of the occluded vein in the BRVO eye (Table 3). VEGF has a vasodilatory effect, and previous reports have suggested that anti-VEGF can be used to suppress the vasodilatory effect and reduce the permeability of the blood vessel [19]. However, in the current study, there was no significant correlation between the number of anti-VEGF injections and vessel diameter, suggesting that anti-VEGF treatment does not suppress vasodilation over long periods. These results suggest that the underlying vasodilation depends on mechanisms other than those mediated by VEGF.

Our study had several limitations. First, although the current study revealed that the RBF reduction was affected by the diameter of, rather than the blood velocity in, the occluded vein in the BRVO eye, we did not provide data pertaining to the longitudinal changes of RBF parameters. Secondly, although we understand BRVO can occur at any location and any diameter of retinal veins, we chose a half-disc diameter away from the optic disc for RBF measurement consistency. Third, as the patients did not always have the fluorescein angiography, the current study could not demonstrate the association between RBF and retinal ischemia. Fourth, we could not investigate the direct relationship between RBF and anti-VEGF treatment in BRVO although there were no correlations between RBF parameters and the number of anti-VEGF injections, and the time from the initial visit. Finally, we recruited a relatively small sample. Therefore, further longitudinal studies with a larger cohort are warranted.

## 5. Conclusions

In BRVO eyes, the RBF and diameter of the occluded vein were significantly lower than those in the non-occluded and the healthy veins. We found that the diameter of the occluded vein was correlated with the BCVA, while neither the time from the initial visit, nor the time from the last injection nor the number of anti-VEGF injections were correlated with the RBF parameters on the occluded vein.

## Figures and Tables

**Figure 1 jcm-09-01847-f001:**
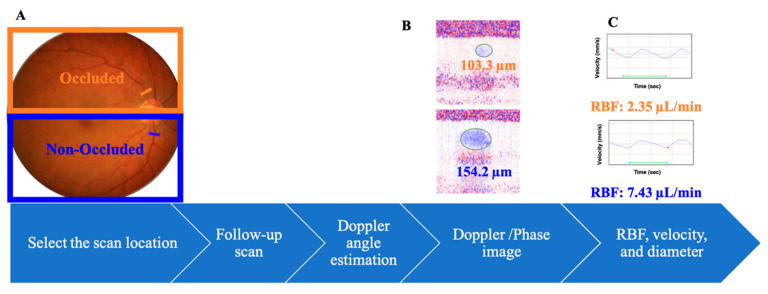
Flow of the image used to measure the retinal blood flow on a Doppler optical coherence tomography flowmeter. (**A**) Color fundus photograph. The bar indicates the measured location on occluded (orange) and non-occluded (blue) veins. (**B**) Phase images with color coding showing the blood flow signals of the vessels on the occluded (orange bar) and non-occluded (blue bar) veins. The numbers indicate the vessel diameter (µm). (**C**) Retinal blood flow (RBF) velocity profiles of the retinal veins within 2 s measured by a Doppler optical coherence tomography (DOCT) flowmeter on the occluded (superior) and non-occluded (inferior) veins; the colored numbers indicate the averaged blood velocity (orange: occluded vein; blue: non-occluded vein).

**Figure 2 jcm-09-01847-f002:**
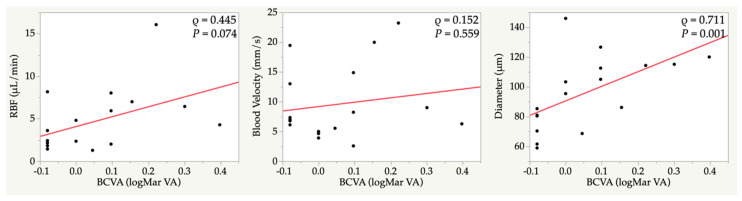
Correlations between the retinal blood flow of the occluded vein and the best corrected visual acuity. RBF, retinal blood flow; BCVA, best corrected visual acuity; ρ, Spearman’s rank correlation coefficient.

**Table 1 jcm-09-01847-t001:** Demographics of the study participants.

	BRVO Eye (*n* = 17)	Fellow Eye (*n* = 17)	*p* Value *
Age (years)	63.1 ± 10.8		-
Male (*n* = 9)	65.6 ± 12.0		
Female (*n* = 8)	60.3 ± 9.3		
Occluded location (superior/inferior)	8/9	-	
Male (*n* = 9)	4/5	-	
Female (*n* = 8)	4/4	-	
BCVA at the initial visit (logMAR)	0.26 ± 0.34	−0.08 ± 0.05	**<0.001**
BCVA (logMAR)	0.06 ± 0.15	−0.09 ± 0.03	**<0.001**
Refractive errors	−0.07 ± 2.45	−0.37 ± 2.12	0.71
IOP (mmHg)	14.9 ± 2.3	14.7 ± 2.1	0.747
MOPP (mmHg)	58.2 ± 8.7	58.4 ± 8.5	0.968
CMT at the initial visit (µm)	439.5 ± 120.0	243.2 ± 19.7	**<0.001**
CMT (µm)	259.0 ± 40.3	244.7 ± 18.5	0.193
Total number of anti-VEGF injections	1.9 ± 1.6	-	
MBP (mmHg)	102.2 ± 13.3		-
HR (bpm)	79.0 ± 15.0		
Time from the initial visit (month)	37.6 ± 32.8		
Time from the last anti-VEGF injection (month)	28.2 ± 21.6		

Significant differences are indicted in bold. * A paired t-test test was used to compare the variables and significance was set at *p* < 0.05. BRVO, branch retinal vein occlusion; BCVA, best-corrected visual acuity; IOP, intraocular pressure; MOPP, mean ocular perfusion pressure; CMT, central macular thickness; VEGF, vascular endothelial growth factor; MBP, mean blood pressure; HR, heart rate. Values are expressed as the mean ± standard deviation (SD).

**Table 2 jcm-09-01847-t002:** Comparison of the retinal blood flow parameters between the BRVO (occluded and non-occluded) and the fellow eyes.

Parameters	Occluded Vein in BRVO Eye (*n* = 17)	Non-Occluded Vein in BRVO Eye (*n* = 17)	Veins in Fellow Eye(*n* = 17)	*p* Value *		
				Occluded vs. Non-Occluded	Occluded vs. Fellow eye	Non-Occluded vs. Fellow eye
RBF (µL/min)	4.7 ± 3.7	10.3 ± 5.1	8.6 ± 4.0	**0.001**	**0.013**	0.828
Blood Velocity (mm/s)	9.6 ± 5.3	12.3 ± 5.0	11.7 ± 5.3	0.105	0.353	0.999
Diameter (µm)	95.9 ± 24.7	127.9 ± 23.7	116.4 ± 13.9	**0.003**	**0.022**	0.317

Significant differences are indicted in bold. * Steel–Dwass test was employed for non-parametric multiple comparisons in RBF parameters among three groups. Significance was set at *p* < 0.05. RBF, retinal blood flow. Values are expressed as the mean ± standard deviation (SD).

**Table 3 jcm-09-01847-t003:** Correlation between the RBF parameters on the occluded vein, and the time from the initial visit, the time from the last anti-VEGF injection and the number of anti-VEGF injections.

	Time from the Initial Visit		Time from the Last Anti-VEGF Injection		Number of Anti-VEGF Injections	
	ρ	*p* Value	ρ	*p* Value	ρ	*p* Value
RBF (µL/min)	−0.158	0.545	−0.188	0.470	0.268	0.298
Blood Velocity (mm/s)	0.286	0.266	0.211	0.416	0.046	0.862
Diameter (µm)	−0.467	0.059	−0.393	0.119	0.279	0.278

RBF, retinal blood flow; VEGF, vascular endothelium growth factor; ρ, Spearman’s rank correlation coefficient.

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
