# Peer review of "Deterioration of Retinal Blood Flow Parameters in Branch Retinal Vein Occlusion Measured by Doppler Optical Coherence Tomography Flowmeter"

_jcm, 2020, doi:10.3390/jcm9061847_

Round 1
Reviewer 1 Report
Dear authors,
I found your work interesting and well written although different points must be improved.
- The inclusion criteria include "acute or chronic BRVO" (line 60). The two types of BRVO present a very different vascular status. The acute phase theoretically displays a blood engorgement upstream the occlusion and lack of blood flow or only partial one (depends on the occlusion) downstream. The chronic phase usually present a thin and weak vessel as the result of the tissue atrophy, or tissue ischemia, or of the formation of a collateral circulation. I suggest to split the two groups and study the two outcomes respectively or to select only one group.
- It is mentioned "RBF was measured at veins half disc diameter away from the optic disc" (line 119), although it was not mentioned the location and the type of the vessel involved by the BRVO. BRVO can affect every type of vessel, a small vessel or a 3-times bigger one, and can occour near to the optic disc or more peripheral; I presume this could severely change RBF parameters. In addition, the localization of the occlusion is important in respect of the macular/foveolar distance, especially when we consider final BCVA.
- Is not mentioned where the DOCT section is made in the occluded vein. Downstream the occlusion I suppose. The section is perpendicular to the vessel walls? The sentence in line 108 is vague, how did you measure vessel diameter? I suggest to add images, in particular images that could help to understand this point.
- Is not clear which are the non-occluded vein in the affected eye that have been chosen.
- Is not clear how many anti-VEGF injections have been employed for each patients, the time interval between the occlusion event and the first IVT, the time interval between the last IVT and the enrollment in the study. If a patient receives the double amount of anti-VEGF compared to another than is easy to expect a different vascular final status. There were PRN injections? Which were the criteria? The macular edema amount were similar in all cases? Have you ever encuntered retinal ischemia, and, in affermative case, how have you treated it? Method section should be improved.
- In the exclusion criteria is mentioned "uncontrolled hypertension" (line 63). There were patient with controlled hypertension or other vascular conditions? Is common to find a patient under antihypertensive treatment which present hypertensive retinopathy or vascular distress for other reasons, maybe past events. These contexts could change retinal vessels characteristics.
- In the Doppler optical coherence tomography section: I found the tecnical doppler functioning excursus interesting and well explained although if you cite the articles [10-13] (line 68) when this technique is already described is not necessary a detailed explanation.
- In Table 1 the BCVA parameters are very good. How was BCVA at presentation? Afterwards is mentioned that "the diameter of the occluded vein was correlated to BCVA". Premising that BCVA values are very good after the injections, the different location of the occlusion and the different vein involved make this correlation even more difficult to establish considering that a vein occlusion far from the macula could not affect the visual acuity at all.
- Is not clear in the conclusions section the relation between the RBF, the diameter of the veins and the blood velocity. I think is very difficult to say, relying on the outcomes of this study, which one affect the other in first place. The word "vasoconstriction" could me misleading.
Good luck with your work
Author Response
JCM-806549
We would like to thank the editor and the reviewers for their time and efforts in reviewing our manuscript and providing their valuable and constructive comments. Please refer to our point-by-point responses listed below.
RESPONSES TO REVIEWER 1:
Comment 1: The inclusion criteria include "acute or chronic BRVO" (line 60). The two types of BRVO present a very different vascular status. The acute phase theoretically displays a blood engorgement upstream the occlusion and lack of blood flow or only partial one (depends on the occlusion) downstream. The chronic phase usually present a thin and weak vessel as the result of the tissue atrophy, or tissue ischemia, or of the formation of a collateral circulation. I suggest to split the two groups and study the two outcomes respectively or to select only one group.
Response 1: Thank you for the reviewer’s comment. We agree with the reviewer’s comment. We excluded the acute cases and only included the patients whose time from the initial visit to the day of DOCT measurement were 6 months or longer. As a result, 17 patients were included in the study, but there were no significant changes in the results (Table 1-3 and Figure 2).
Comment 2: It is mentioned "RBF was measured at veins half disc diameter away from the optic disc" (line 119), although it was not mentioned the location and the type of the vessel involved by the BRVO. BRVO can affect every type of vessel, a small vessel or a 3-times bigger one, and can occour near to the optic disc or more peripheral; I presume this could severely change RBF parameters. In addition, the localization of the occlusion is important in respect of the macular/foveolar distance, especially when we consider final BCVA.
Response 2: Thank you for the reviewer’s comment, and we agree with this reviewer’s concern. We have added that “Secondly, although we understand BRVO can occur any location and any diameter of retinal veins we have chosen a half-disc diameter away from the optic disc for RBF measurement consistency” (line 209-210).
Comment 3: Is not mentioned where the DOCT section is made in the occluded vein. Downstream the occlusion I suppose. The section is perpendicular to the vessel walls? The sentence in line 108 is vague, how did you measure vessel diameter? I suggest to add images, in particular images that could help to understand this point.
Response 3: We appreciate the reviewer’s comment. DOCT measures the horizontal diameter of the vessel based on the phase image. We have added a more detailed explanation regarding the way to measure vessel diameters (line 110-112). Additionally, we have revised Figure 1-B to help readers understand how to measure vessel diameter.
Comment 4: Is not clear which are the non-occluded vein in the affected eye that have been chosen.
Response 4: Thank you for the comment. We have elaborated on this and revised the explanation as follows “2. A non-occluded vein on the opposite hemisphere of the occluded vein in BRVO eye” (line 123-124).
Comment 5: Is not clear how many anti-VEGF injections have been employed for each patients, the time interval between the occlusion event and the first IVT, the time interval between the last IVT and the enrollment in the study. If a patient receives the double amount of anti-VEGF compared to another than is easy to expect a different vascular final status. There were PRN injections? Which were the criteria? The macular edema amount were similar in all cases? Have you ever encuntered retinal ischemia, and, in affermative case, how have you treated it? Method section should be improved.
Response 5: Thank you very much for the insightful comments. We agree with the reviewer’s comment so the presence of retinal ischemia, different number of VEGF injections, and different time interval of the IVT might produce different RBF results. To clarify, we have added the comparison between the time from the last IVT and the enrollment (the day of the DOCT measurement) in the study (Table 3). Additionally, the information of the central macular thickness at the initial visit and on the day of DOCT flowmeter measurement, the total number of IVT, and the time from the initial visit where all patients underwent an IVT were added in Table 1. However, our data did not show the direct relationship between RBF and anti-VEGF treatment in BRVO and not all the patients underwent the fluorescein angiography. Therefore, we have added these as limitations (line 210-214). Also, as the reviewer suggested, we have added the diagnostic criteria of BRVO and its treatment regiment in the Methods section (line 65-69).
Comment 6: In the exclusion criteria is mentioned "uncontrolled hypertension" (line 63). There were patient with controlled hypertension or other vascular conditions? Is common to find a patient under antihypertensive treatment which present hypertensive retinopathy or vascular distress for other reasons, maybe past events. These contexts could change retinal vessels characteristics.
Response 6: Thank you for the comment. As we assumed the extremely high blood pressure might have affected RBF, we excluded patients who had extremely high blood pressure (>blood pressure >160/100 mmHg) on the day of the DOCT measurement. Additionally, we have excluded patients with any history and evidence of past events, or the presence of hypertensive retinopathy or vascular distress. We have added this information as the exclusion criteria (line 69-72).
Comment 7: In the Doppler optical coherence tomography section: I found the tecnical doppler functioning excursus interesting and well explained although if you cite the articles [10-13] (line 68) when this technique is already described is not necessary a detailed explanation.
Response 7: We agree with the reviewer’s comment, and therefore we summarized the explanation of the Doppler optical coherence tomography (line 90-112).
Comment 8: In Table 1 the BCVA parameters are very good. How was BCVA at presentation? Afterwards is mentioned that "the diameter of the occluded vein was correlated to BCVA". Premising that BCVA values are very good after the injections, the different location of the occlusion and the different vein involved make this correlation even more difficult to establish considering that a vein occlusion far from the macula could not affect the visual acuity at all.
Response 8: Thank you for the thoughtful comment. In agreement with the reviewer’s suggestion, we have added the BCVA data at the initial visit in Table 1. The BCVA at the initial visit (before the anti-VEGF treatment) was not good (logMAR 0.26 ± 0.34) most likely due to ME (mean CMT at the initial visit was 439.5µm).
Comment 9: Is not clear in the conclusions section the relation between the RBF, the diameter of the veins and the blood velocity. I think is very difficult to say, relying on the outcomes of this study, which one affect the other in first place. The word "vasoconstriction" could me misleading.
Response 9: I agree with the author’s comment. As pointed out, it would be difficult to say which one affects the other as conclusions. Also, we agree that the word “vasoconstriction” is not appropriate to use in our study. Therefore, we have revised the conclusions accordingly (line 30-34 and 218-222). We appreciate the valuable comments.
Again, thank you very much for giving us the opportunity to strengthen our manuscript with your valuable comments and queries. We hope that these revisions persuade the editor and the reviewers to accept our submission.
Reviewer 2 Report
This is an interesting study using DOCT to document blood flow in branch retinal vein occlusion. The results show that the flow in the occluded retinal vein is abnormally reduced. The vessel diameter is correlated with BCVA.
I suggest, to improve the quality of presentation, to show a scatterplot of vessel diameter and flow versus BCVA.
Minor spell check is required.
Author Response
JCM-806549
We would like to thank the editor and the reviewers for their time and efforts in reviewing our manuscript and providing their valuable and constructive comments. Please refer to our point-by-point responses listed below.
RESPONSES TO REVIEWER 2:
Comment 1: I suggest, to improve the quality of presentation, to show a scatterplot of vessel diameter and flow versus BCVA.
Response 1: Thank you very much for your thoughtful suggestion. As suggested, we have revised Table 3 to scatterplots in Figure 2.
Comment 2: Minor spell check is required.
Response 2: We appreciate the comment. We have rechecked the spell throughout the manuscript.
Again, thank you very much for giving us the opportunity to strengthen our manuscript with your valuable comments and queries. We hope that these revisions persuade the editor and the reviewers to accept our submission.
Reviewer 3 Report
Takahashi et al is a well-written article that could be of interest to many readers but is short of delivering a complete recount of the study design and aim. The abstract is clear, the introduction is informative but the methodology and results are quite difficult to follow and the discussion employs vocabulary not related to the findings of the investigation. I find that many important details of the study are missing.
Some of my comments are below:
Abstract: Nicely presented but there are two immediate questions: 1) is the healthy vein assumed to be the one in the contralateral eye? What would be the reasons for a healthy vessel diameter to be intermediate in size? 2) the conclusion mentions that there is correlation between vein size and VA but there is no indication of this in the abstract results. Small adjustments to the presentation of the information can solve this issue. However, after reading the rest of the manuscript, I think there is considerable information missing about the VEGF participants, which should also be included in the abstract.
The inclusion/exclusion criteria reports on the average age of the nine males in this study. What was the average age of the 11 females? What was the specialist’s criteria for the diagnosis of BRVO? This has to be included.
The exclusion criteria does no mention pre-existing retinal dystrophies. Please add this to the inclusion/exclusion list if appropriate.
Please rephrase this statement for clarity: ‘who need the anti-VEGF injection due to macular edema on the examination day were excluded’. All the information I have about VEGF treatment of participants is in the title of table 4. This group of participants need to be described in the inclusion criteria, and all the details of frequency of injection and number of injections that table 4 alludes to. How many participants were included in the analysis? What are their specific vein parameters and VA values?
Figure 1 legend mentions ‘arrowheads’ but they are not shown in figure 1B.
The method section does not indicate was VA chart was used in testing participants. Table 1 LogMAR is confusing for the general reader and better not to use terms like higher/lower. 0.09 (6/7.5+) is worse than -0.08 (6/5-). The division into male/female could be clearer if the occluded vein location is also reported by gender. Average female age vs male average age should be reported in table 1.
A paired t-test analysis of BRVO eyes with fellow eyes is OK but you need to name the test. A t-test analysis of the three types of veins per eye is not correct; you need to use a multiple comparisons test. In all cases, tell the reader the name of the test you used.
Page 4, line 145-146 report no significant differences in blood velocity among the veins (occluded vein, non-occluded vein, vein in fellow eyes, P > 0.05). I do not understand how you got this p value using a t-test (paired comparison). Please revise your statistical analysis.
Lines 148- 149 describing the diameter of veins as ‘intermediate’ is not correct. The statistical value compared with the occluded and non occluded veins need to be shown.
Table 2 calls the contralateral eye ‘normal’. Given the inclusion criteria, I suggest you refrain from assuming normality of the non-affected eye.
Table 3 does not help me understand the correlation. I would expected this table to show the data for each of the 20 participants versus their VA. Table 3 also shows the ratios calculated by dividing the variables of the occluded vein by the variables of the non-occluded vein. It is not clear to me what is the significance of showing a ratio. This needs to be explained in the methods.
The discussion implies that BRVO prognosis can be inferred from the diameter of the occluded vein given that it correlated with the BCVA. I do not understand how ‘prognosis’ can be inferred from this. This study did not evaluate patients as a function of time. If they did, this is not clear.
Lines 193-195 The discussion correctly mentions that anti-VEGF injection causes vasoconstriction of the retinal vein, but I disagree that there is correlation with the BCVA.
I disagree with the last sentence in the conclusion referring to this study providing new insight into the pathogenesis of BRVO. The development of the disease was not studied here.
Author Response
JCM-806549
We would like to thank the editor and the reviewers for their time and efforts in reviewing our manuscript and providing their valuable and constructive comments. Please refer to our point-by-point responses listed below.
RESPONSES TO REVIEWER 3:
Comment 1: Abstract: Nicely presented but there are two immediate questions: 1) is the healthy vein assumed to be the one in the contralateral eye? What would be the reasons for a healthy vessel diameter to be intermediate in size?
Comment 10: Lines 148- 149 describing the diameter of veins as ‘intermediate’ is not correct. The statistical value compared with the occluded and non occluded veins need to be shown.
Response 1 &10: Thank you very much for the comments. For the revised analysis, Steel-Dewas test was employed for the comparisons in RBF parameters between BRVO and fellow eyes, and we found that there was no significant difference in the diameter between the non-occluded vein and the vein in fellow eyes (Table 2).
Comment 2: 2) the conclusion mentions that there is correlation between vein size and VA but there is no indication of this in the abstract results. Small adjustments to the presentation of the information can solve this issue. However, after reading the rest of the manuscript, I think there is considerable information missing about the VEGF participants, which should also be included in the abstract.
Response 2: Thank you for the thoughtful suggestion. As suggested, we have added the information about the correlation between the diameter of the vein and BCVA in the abstract result (line 27-29). Additionally, we included that there were no significant differences between RBF parameters and the time from the initial visit as well as the total number of anti-VEGF injections (line 29-30).
Comment 3: The inclusion/exclusion criteria reports on the average age of the nine males in this study. What was the average age of the 11 females? What was the specialist’s criteria for the diagnosis of BRVO? This has to be included.
Response 3: Thank you for the comments. As the author suggested, we have added information about the average ages by gender in Table 1. We have also added the diagnosis criteria of BRVO in the Methods section (line 65-67).
Comment 4: The exclusion criteria does no mention pre-existing retinal dystrophies. Please add this to the inclusion/exclusion list if appropriate.
Response 4: Thank you for the valuable suggestion. We have added the pre-existing retinal dystrophies as one of the exclusion criteria (line 70).
Comment 5: Please rephrase this statement for clarity: ‘who need the anti-VEGF injection due to macular edema on the examination day were excluded’. All the information I have about VEGF treatment of participants is in the title of table 4. This group of participants need to be described in the inclusion criteria, and all the details of frequency of injection and number of injections that table 4 alludes to. How many participants were included in the analysis? What are their specific vein parameters and VA values?
Response 5: We agree with the reviewer’s comment. Therefore, we rephrased the statement to “We also excluded the patients who underwent the anti-VEGF injection due to ME on the day of the DOCT-flowmeter measurement” (line 72-74). Also, we have added the information of the 17 participants about the total number of anti-VEGF injections and the time from the initial visit in Table 1. Additionally, we have analyzed the correlations between the vein parameters (RBF parameters) and the BCVA in Figure 2.
Comment 6: Figure 1 legend mentions ‘arrowheads’ but they are not shown in figure 1B.
Response 6: Thank you for the heads-up. We have corrected ‘arrowheads’ to ‘bar’ (line 85).
Comment 7: The method section does not indicate was VA chart was used in testing participants. Table 1 LogMAR is confusing for the general reader and better not to use terms like higher/lower. 0.09 (6/7.5+) is worse than -0.08 (6/5-). The division into male/female could be clearer if the occluded vein location is also reported by gender. Average female age vs male average age should be reported in table 1.
Response 7: Thank you for the thoughtful comment. Following the feedback, we have added the information that “a standard logarithmic visual acuity chart was used” (line 117-118) and amended to use the expression “worse” instead of higher (line 139). Also, we have provided information about the occluded vein locations and the mean ages by gender in Table 1.
Comment 8: A paired t-test analysis of BRVO eyes with fellow eyes is OK but you need to name the test. A t-test analysis of the three types of veins per eye is not correct; you need to use a multiple comparisons test. In all cases, tell the reader the name of the test you used.
Response 8: Thank you very much for the valuable comments. We agree with the reviewer’s comment. We reconsidered the stats, so the Steel-Dewas test was employed for multiple comparisons, especially to examine the differences in RBF parameters between the BRVO and the fellow eyes in Table 2. Also, as suggested, we indicate the name of the stats to be used in all cases.
Comment 9: Page 4, line 145-146 report no significant differences in blood velocity among the veins (occluded vein, non-occluded vein, vein in fellow eyes, P > 0.05). I do not understand how you got this p value using a t-test (paired comparison). Please revise your statistical analysis.
Response 9: Thank you for the comment. As the reviewer suggested, we reanalyzed it again, and there were no differences in blood velocity among the veins in Table 2 (Please also refer to the dot plot separately sent).
Comment 11: Table 2 calls the contralateral eye ‘normal’. Given the inclusion criteria, I suggest you refrain from assuming normality of the non-affected eye.
Response 11: We agree with the comment, and therefore we have corrected ‘normal eye’ to ‘fellow eye’ in Table 2.
Comment 12: Table 3 does not help me understand the correlation. I would expected this table to show the data for each of the 20 participants versus their VA. Table 3 also shows the ratios calculated by dividing the variables of the occluded vein by the variables of the non-occluded vein. It is not clear to me what is the significance of showing a ratio. This needs to be explained in the methods.
Response 12: Thank you for the thoughtful comments. We agree and changed Table 3 to the scatterplots as Figure 2. Also, we agree with the reviewer’s comment that the ratios RBF parameters are confusing. Thus, we have removed the sentences mentioned the ratio from the manuscript.
Comment 13: The discussion implies that BRVO prognosis can be inferred from the diameter of the occluded vein given that it correlated with the BCVA. I do not understand how ‘prognosis’ can be inferred from this. This study did not evaluate patients as a function of time. If they did, this is not clear.
Response 13: We agree with the reviewer's comment, and we have removed the sentence from the discussion.
Comment 14: Lines 193-195 The discussion correctly mentions that anti-VEGF injection causes vasoconstriction of the retinal vein, but I disagree that there is correlation with the BCVA.
Response 14: Thank you very much for the comments. We agree with the comment, and thus we have removed “there is a correlation between the anti-VEGF injections and the BCVA” from the discussion.
Comment 15: I disagree with the last sentence in the conclusion referring to this study providing new insight into the pathogenesis of BRVO. The development of the disease was not studied here.
Response 15: We agree with the reviewer’s comment. As suggested, it was not reasonable to conclude; therefore, we have removed the last sentence in conclusion.
Again, thank you very much for giving us the opportunity to strengthen our manuscript with your valuable comments and queries. We hope that these revisions persuade the editor and the reviewers to accept our submission.

Round 2
Reviewer 1 Report
Good work! I found a remarkable improvement in the reading of this second version of your manuscript.
A little adjustment is needed in the abstract: the number of patients is still 20
Author Response
JCM-806549
We would like to thank the editor and the reviewer 1 for their time and efforts in reviewing our manuscript and providing their valuable and constructive comment. Please refer to our point-by-point responses listed below.
RESPONSE TO REVIEWER 1:
Comment 1: A little adjustment is needed in the abstract: the number of patients is still 20
Response 1: Thank you for the reviewer’s comment. As suggested, we have revised the number of patients from 20 to 17 in the abstract (Line 15).
Again, thank you very much for giving us the opportunity to strengthen our manuscript with your valuable comment. We hope that these revisions persuade the editor and the reviewers to accept our submission.